# Exogenous Melatonin Enhances Dihydrochalcone Accumulation in *Lithocarpus litseifolius* Leaves via Regulating Hormonal Crosstalk and Transcriptional Profiling

**DOI:** 10.3390/ijms25094592

**Published:** 2024-04-23

**Authors:** Wenlong Zhang, Yuqi Sun, Hongfeng Wang, Mingfeng Xu, Chunmei He, Congcong Wang, Yongli Yu, Zongshen Zhang, Lingye Su

**Affiliations:** 1School of Biology Engineering, Dalian Polytechnic University, Dalian 116034, China; 210210710000167@xy.dlpu.edu.cn (W.Z.); 220210710000166@xy.dlpu.edu.cn (Y.S.); yuyongli203@outlook.com (Y.Y.); 2Guangdong Academy of Forestry, Guangdong Provincial Key Laboratory of Silviculture Protection and Utilization, Guangzhou 510520, China; wanghf@sinogaf.cn (H.W.); xumingfeng@sinogaf.cn (M.X.); meizi@sinogaf.cn (C.H.); wangcc@sinogaf.cn (C.W.)

**Keywords:** antioxidant, elicitor, flavonoid, metabolic pathway regulation, phlorizin, trilobatin

## Abstract

Dihydrochalcones (DHCs) constitute a specific class of flavonoids widely known for their various health-related advantages. Melatonin (MLT) has received attention worldwide as a master regulator in plants, but its roles in DHC accumulation remain unclear. Herein, the elicitation impacts of MLT on DHC biosynthesis were examined in *Lithocarpus litseifolius*, a valuable medicinal plant famous for its sweet flavor and anti-diabetes effect. Compared to the control, the foliar application of MLT significantly increased total flavonoid and DHC (phlorizin, trilobatin, and phloretin) levels in *L. litseifolius* leaves, especially when 100 μM MLT was utilized for 14 days. Moreover, antioxidant enzyme activities were boosted after MLT treatments, resulting in a decrease in the levels of intracellular reactive oxygen species. Remarkably, MLT triggered the biosynthesis of numerous phytohormones linked to secondary metabolism (salicylic acid, methyl jasmonic acid (MeJA), and ethylene), while reducing free JA contents in *L. litseifolius*. Additionally, the flavonoid biosynthetic enzyme activities were enhanced by the MLT in leaves. Multiple differentially expressed genes (DEGs) in RNA-seq might play a crucial role in MLT-elicited pathways, particularly those associated with the antioxidant system (SOD, CAT, and POD), transcription factor regulation (*MYBs* and *bHLHs*), and DHC metabolism (*4CL*, *C4H*, *UGT71K1*, and *UGT88A1*). As a result, MLT enhanced DHC accumulation in *L. litseifolius* leaves, primarily by modulating the antioxidant activity and co-regulating the physiological, hormonal, and transcriptional pathways of DHC metabolism.

## 1. Introduction

*Lithocarpus litseifolius*, an evergreen Fagaceae arbor, has long been extensively utilized as a beneficial medicinal plant in Southern China [1]. The leaves of *L. litseifolius* (sweet tea) exhibit various preventive and therapeutic properties, such as anti-diabetes, anti-hypertension, and anti-hyperlipidemia effects [1]. Previous pharmacological analyses have revealed that dihydrochalcones (DHCs), falling into a specific class of flavonoids, are responsible for their unique sweet flavors and health benefits [2]. DHC is generally produced by specific plants. The majority of them (i.e., apples) mainly produce only phlorizin (phloretin 2′-*O*-glucoside), whereas *L. litseifolius* can synthesize both phlorizin and its isomer trilobatin (phloretin 4′-*O*-glucoside) in large quantities [1]. Phloridzin is a precursor to the anti-diabetes drug Dapagliflozin [3], while trilobatin serves as a natural low-calorie sweetener [4]. As a result, *L. litseifolius* offers an excellent resource of diversified high-value DHCs and has been approved as a new food ingredient in China with considerable market demand [1]. Nevertheless, wild *L. litseifolius* is typically dispersed across the gully and hillside, resulting in picking difficulty and quantity restrictiveness [1]. Additionally, the cultivation of seedlings generally accumulates limited amounts of DHCs, making it challenging to meet the demand for large-scale markets [5]. Therefore, it is essential to explore how to improve the yield of DHCs in *L. litseifolius* cultivation.

Among the multiple methods of regulatory management, elicitation strategies represent one of the most practically effective methods for enhancing metabolite production [6]. As a multi-functional biomolecule, melatonin (*N*-acetyl-5-methoxytryptamine; MLT) has received attention worldwide for its diverse co-regulatory roles in plant growth, development, and stress response [7]. Notably, MLT can be utilized as a novel simultaneous elicitor to promote flavonoid biosynthesis [7]. For instance, exogenous MLT increased isoflavone levels in UV-treated soybean sprouts, and stimulated the accumulation of anthocyanin, flavonols, and proanthocyanins in crabapple leaves [8,9]. However, little is known about the regulatory roles of MLT in DHC biosynthesis, and its physiological and molecular mechanisms in *L. litseifolius* leaves still require further research.

MLT acts primarily as an antioxidant to protect cells from the oxidative stress triggered by excessive levels of reactive oxygen species (ROS) [10]. Multiple ROS scavenging enzymes, including glutathione reductase (GR), superoxide dismutase (SOD), catalase (CAT), and peroxidase (POD), are initially activated in the process of MLT-induced flavonoid biosynthesis [11]. For instance, exogenous MLT improved anthocyanin accumulation by boosting the activities of SOD, CAT, and POD in cabbage [12]. Moreover, the multiple regulatory roles of MLT are heavily dependent on the crosstalk between it and other phytohormones [13]. MLT could stimulate various defense-related hormones, including MeJA, SA, and ethylene (ET), resulting in the potential to enhance plant secondary metabolism [13]. Despite numerous studies conducted on MLT-mediated hormonal crosstalk, it remains essential to conduct further study on the interactions between multiple phytohormone forms and MLT-induced flavonoid biosynthesis.

A complex transcriptional network is triggered following the MLT-induced hormonal signaling [14]. Various transcription factors (TFs), such as MYB, bHLH, WD40, and WRKY families, play a crucial role in transcriptional regulation in response to MLT [12,15]. These transcription factors may directly influence the expression levels of downstream genes involved in flavonoid metabolism and plant growth [16]. For instance, *LjaMYB12* from *Lonicera japonica* positively regulates the expression of genes in flavonoid biosynthetic pathways, such as phenylalanine ammonia-lyase gene (*PAL*), cinnamate 4-hydroxylase gene (*C4H*), and chalcone synthase gene (*CHS*) in transgenic *Arabidopsis*, leading to an increase in PAL activity and flavonoid content [17].

In this study, we aimed to investigate the regulatory mechanisms of MLT on DHC biosynthesis in *L. litseifolius* leaves for the first time, as far as we know. Furthermore, comprehensive research was conducted on the MLT-induced changes in antioxidant capacity (H_2_O_2_, malondialdehyde (MDA), GR, SOD, POD, and CAT), phytohormone crosstalk (SA, JA, and ET classes), and transcriptional profiling (associated TFs and structural genes) (Figure 1). It was found that MLT could act as a potential elicitor of DHC accumulation in *L. litseifolius* leaves, primarily by increasing the antioxidant activity and by co-regulating the physiological, hormonal, and transcriptional processes of DHC metabolism.

## 2. Results

### 2.1. Effect of MLT Treatment on DHC Biosynthesis

To explore the impact of MLT on DHC biosynthesis in *L. litseifilus*, the seedlings were foliarly exposed to various MLT concentrations (0, 50, 100, and 200 μM), and the total flavonoid (TTF) levels were initially determined. Compared to CK, TTF yields were significantly enhanced following 14 days of MLT treatments at 100 and 200 μM, respectively (Figure 2a). Among them, the leaf samples treated with 100 μM MLT accumulated the highest TTF contents, which increased by 39.60% (20.41 ± 1.72 mg g^−1^ DW) above CK (Figure 2a).

Furthermore, the high-performance liquid chromatography (HPLC) system was applied to determine the yield of three representative DHCs (phlorizin, trilobatin, and phloretin) in *L. litseifilus* under the context of 50, 100, and 200 μM MLT treatments. No obvious differences between the three DHC levels were detected in the 50 or 200 μM MLT-treated groups after 14 days compared to CK (except for trilobatin at 50 μM MLT) (Figure 2b–d and Appendix A). However, it was discovered that exogenous MLT at 100 μM elicited the highest levels of phlorizin (10.12 ± 2.45 mg g^−1^ DW), trilobatin (5.04 ± 0.72 mg g^−1^ DW), and phloretin (0.53 ± 0.06 mg g^−1^ DW), showing the enhancement by 130.37%, 65.44%, and 34.70% relative to CK, respectively (Figure 2b–d). In general, MLT treatments enhanced DHC levels in *L. litseifilus*, with 100 μM as the optimal MLT-elicited concentration.

### 2.2. Transcriptome Sequencing and DEG Pathway Analysis

To elucidate the transcriptional regulatory network in MLT-elicited *L. litseifilus* seedlings, six RNA-seq samples were collected from the treatment groups of water (CK) and 100 μM MLT (MLT 100) after 14 days and then sequenced using the Illumina Hiseq platform. A total of 71.18 Gb raw data and 69.77 Gb clean data were generated, with Q20 and Q30 bases exceeding 96.7% and 91.1% (Appendix A). The *L. litseifilus* transcriptomes contained 20,381 unigenes after mapping and statistics. These unigenes had an average length of 2381 bp, and the N50 value achieved a length of 2692 bp. Among them, 20,114 unigenes were effectively annotated in at least one of the public databases. Among them, 10,095 (50.2%) were annotated in all seven databases (Appendix A).

The DEG information was further analyzed by comparing MLT 100 with CK. A total of 2864 DEGs were identified based on the criteria (|log_2_ fold change| > 0 and *p* < 0.05). The biological roles of these DEGs were investigated through KEGG and GO pathway enrichment analyses. The ‘photosynthesis’ showed the most enriched KEGG pathway, followed by ‘flavone and flavonol biosynthesis’ and ‘sulfur relay system’ (Figure 3, Appendix A). Interestingly, the largest number of KEGG items were associated with secondary metabolism, while ‘glucosinolate biosynthesis’, ‘biosynthesis of various plant secondary metabolites’, and ‘flavonoid biosynthesis’ were also listed in the top 20 enriched KEGG pathways of DEGs. (Figure 3, Appendix A). Furthermore, the analysis of GO also revealed that the ‘metabolic process’ category had the highest number of differentially expressed genes (DEGs) in the aforementioned comparison (Appendix A).

### 2.3. Effects of MLT on the Antioxidant System and Related Gene Expression

To investigate the putative effects of MLT on the antioxidant system in *L. litseifilus* seedlings, the intracellular ROS content (H_2_O_2_), lipid superoxidation level (MDA), and antioxidant enzyme activities (GR, SOD, CAT, and POD) were determined in the plants treated with MLT (0, 50, 100, and 200 μM). Only 100 μM MLT treatments significantly reduced the H_2_O_2_ generation in leaves by 27.63% (4.70 ± 0.18 mmol g^−1^ protein) as compared to CK (Figure 4a). Furthermore, a positive relationship was discovered between MDA levels and MLT concentrations (Figure 4b). The MDA contents were significantly lower in MLT-treated plants at 50 and 100 μM, down by 28.95% (48.06 ± 7.73 mmol g^−1^ protein) and 22.99% (38.17 ± 6.47 mmol g^−1^ protein) compared to CK, respectively (Figure 4b).

Furthermore, it was found out that exogenous MLT at all doses (50, 100, and 200 μM) significantly activated all four antioxidant enzymes (GR, SOD, CAT, and POD), with an improvement by 16.09, 2.16, 8.50, and 1.84 times compared to the control group (CK), respectively (Figure 4c–f). In the RNA-seq transcriptome, a total of 3, 9, 10, and 57 unigenes were, respectively, identified as *GRs*, *SODs*, *CATs*, and *PODs* (Figure 4g, Appendix A). Among them, one *SOD*, one *CAT*, and six *PODs* were up-regulated in the comparison between MLT 100 and K (Figure 4g). When compared to CK, the maximum enhancement in expression of *SOD* (42812/f3p0/887), *CAT* (29712/f5p0/1810), and *POD* (18736/f4p0/2391) was 1.51 times, 6.33 times, and 6.21 times, respectively (Figure 4g, Appendix A).

### 2.4. Effects of MLT on Phytohormone Crosstalks and Related Gene Expression

The crosstalk relationships between MLT and several phytohormones in regulating DHC biosynthesis were explored by determining the amounts of hormones (e.g., two types of SA, seven metabolites involved in JA biosynthesis, and 1-aminocyclopropane carboxylic acid (ACC, ET precursor)) in CK and 100 μM MLT-treated groups (the optimal group for DHC elicitation). Compared to CK, the free SA level increased by 34.24% in *L. litseifilus* leaves, while SA *β*-glucoside (SAG) amount declined by 18.25% under the context of MLT treatment (Figure 5, Appendix A). Although the lower levels of free JA and two JA conjugate types (JA-Ile and JA-Val) occurred due to MLT treatments, MeJA and 12-oxophytodienoate (OPDA, JA precursor) content was, respectively, improved by 74.02% and 26.89% compared to that of CK (Figure 5, Appendix A). Furthermore, MLT improved the biosynthesis of ACC, showing an enhancement by 28.56% (Figure 5, Appendix A).

Subsequently, the DEGs associated with the biosynthesis and signaling pathways of these phytohormones were examined. A total of twenty-one DEGs in the comparison between MLT 100 and CK were associated with hormone biosynthesis, including three for SA, three for JA, and fifteen for ETH (Appendix A). Furthermore, it was discovered that 204 unigenes in the KEGG database were categorized as ‘phytohormone signaling’. The MLT-induced expression of *AUX1* (25295/f3p0/2023), *JAZ* (38589/f3p0/1249), and *ERF105* (34493/f9p0/1534) was at least 1.5 times higher than in CK (Appendix A).

### 2.5. Effects of MLT on TF Families

Multiple TF families play a critical role in the biological processes of plants, such as secondary metabolism [18]. According to the RNA-seq results, 1155 TF unigenes belonging to 29 families were identified in *L. litseifilus*, of which 209 TF unigenes were up-regulated by MLT (Appendix A). Four major classes of TFs, including MYB, bHLH, WD40, and WRKY, were mainly involved in the regulation of flavonoid metabolism [16]. MLT up-regulated nineteen, eight, seven, and eight unigenes from MYB, bHLH, WD40, and WRKY families, respectively (Figure 6, Appendix A). These DEGs mainly included MYB (*MYB3R1*, *MYB1R1*, *MYB TT2*, *MYB4*, and *MYB36*), bHLH (*bHLH130*, *bHLH128*, *bHLH93*, *bHLH79*, *bHLH35*, and *PIF3*), WD40 (*AN11*, *WDR76*, *WDR43*, *WDR26*, and *WDR20*), and WRKY (*WRKY24*, *WRKY70*, *WRKY7*, *WRKY2*, *WRKY72*, *WRKY20*, *WRKY3*, and *WRKY31*) (Figure 6).

### 2.6. Effects of MLT on DHC Biosynthesis and Related Gene Expression

Multiple biosynthetic enzymes catalyze the MLT-elicited flavonoid metabolism stepwise [19]. Therefore, the activities of three key enzymes in the phenylalanine pathway (PAL, C4H, and 4-coumaric acid CoA ligase (4CL)) were initially measured following MLT elicitation (50, 100, and 200 μM). As shown in Figure 7, the PAL and C4H activities were only triggered by 100 μM MLT treatment, while all MLT groups significantly increased the 4CL activities in *L. litseifilus*. Notably, the highest MLT-induced enhancement of all enzyme activities occurred at 100 μM, up by 45.85%, 89.13%, and 822.73% for PAL, C4H, and 4CL, respectively (Figure 7).

The candidate critical genes involved in the DHC biosynthesis pathway were then screened by KEGG analysis. A total of 50 up-regulated DEGs were discovered in the flavonoid biosynthesis-associated KEGG categories (‘flavone and flavonol biosynthesis’, ‘flavonoid biosynthesis’, and ‘phenylpropanoid biosynthesis’) (Appendix A). Among them, seven DEGs involved in flavonoid skeleton biosynthesis showed an increase in expression by at least 1.5 times in MLT-treated groups, including one *PAL* (15514/f17p0/2519), two *C4Hs* (27236/f3p0/1923 and 27875/f12p0/1831) and four *4CLs* (34202/f4p0/1578, 5232/f2p0/2628, 34871/f3p0/1566, and 26318/f3p0/1973) (Figure 7). Additionally, 98 *UGT* family genes were identified in RNA-seq data, 20 of which showed the MLT-induced increase by at least 1.5 times over CK (Appendix A). *UGT91A1* (30673/f2p0/1775) exhibited the most significant change of expression in MLT-treated groups, while the level of *UGT71K1* (39345/f2p0/1284 and 19493/f4p0/2347) and *UGT88A1* (32492/f35p0/1640) expression was enhanced by 2.39, 2.42, and 1.53 times, respectively (Figure 7, Appendix A). The UGT families, as mentioned above, have been linked to the biosynthesis of flavonol and DHC glycosides [4,20,21]. Therefore, it was suggested that *PAL*, *C4Hs*, *4CLs*, *UGT91A1*, *UGT71K1*, and *UGT88A1* might be crucial to the transcriptional regulation of MLT-elicited DHC biosynthesis in *L. litseifilus*.

### 2.7. Quantitative Real-Time PCR (qRT–PCR) Validation of DEGs

Nine up-regulated DEGs (*PAL*, *4CL*, *UGT*, *SOD*, *POD*, *MYB TT2*, *bHLH130*, *SGT1*, and *JAZ*) were selected for qRT-PCR analysis to validate the RNA-Seq findings. The expression levels of most genes exhibited a high degree of concordance with the RNA-seq findings (Figure 8). Hence, the RNA-seq data might be useful to investigate the molecular mechanism of MLT-elicited DHC production in *L. litseifilus*.

### 2.8. Correlation Analysis

Principal component analysis (PCA) was further conducted to explore the correlation among the multiple physiological indexes. As shown in Figure 9, significant divisions occurred among three MLT treatment groups (MLT-50, MLT-100, and MLT-200) and CK, indicating that the effects of MLT elicitation in *L. litseifilus* were typically dose-dependent. Furthermore, the first (PC1) and second (PC2) principal components accounted for 72.9% and 11.3% of total variability, respectively (Figure 9). H_2_O_2_ and MDA were both negatively connected with PC1. In contrast, the other indicators were all positively correlated with PC1 (Figure 9). These variations were primarily associated with phlorizin, trilobatin, and C4H values (Figure 9).

Furthermore, an investigation was conducted into the correlation between 42 MLT-induced TFs and their ten potential downstream genes involved in DHC biosynthesis (Figure 10). Over 45% of TF genes, including *MYB4* and *bHLH35*, showed a significant level of co-expression (*p* < 0.05) with *4CL* (34202/f4p0/1578), *C4H* (27236/f3p0/1923), *UGT71K1* (39345/f2p0/1284), and *UGT88A1* (32492/f35p0/1640) (Figure 10).

## 3. Discussion

### 3.1. MLT Elicitation of DHC Levels

Flavonoids are polyphenolic chemicals widely found in plants and serve as the major bioactive secondary metabolites [19]. As a category of rare flavonoids, DHCs may be generally produced in restricted quantities by specific plants, particularly in *Malus* and *Lithocarpus* species [2]. Although the DHCs in *L. litseifolius* have gained popularity worldwide for diabetic therapy [1], little is known about their elicitation strategies and regulatory mechanisms in plants. In [22], it was found out that green light treatment increased PAL activity and phlorizin accumulation in *L. litseifolius*. However, this strategy is costly and ineffective for large-scale production. The role of MLT as an effective and economical regulator in eliciting plant metabolite synthesis has been extensively researched [7]. The treatment with 10 µM MLT resulted in the maximum campesterol and stigmasterol production in *Lemna aequinoctialis* cultures, while 200 µM MLT dramatically improved flavonol and flavanol biosynthesis in grape berries [23,24]. These findings indicated that the effects of MLT elicitation were dose- and species-dependent. Herein, the influence of MLT on the level of DHC in the leaves of *L. litseifolius* was thoroughly investigated (Figure 2 and Appendix A). Compared to CK, foliarly applied MLT significantly enhanced the levels of TTF, phlorizin, trilobatin, and phloretin, by up to 39.60%, 130.37%, 65.44%, and 34.70%, respectively (Figure 2). Interestingly, the MLT 100 group showed the highest levels of all the aforementioned indexes (Figure 2), indicating that the role of MLT on DHC biosynthesis was also substantially dose-dependent, which was concurrent with the study of Chang et al. (2023) [25]. Consequently, MLT elicitation provided a viable strategy for enhancing DHC accumulation in *L. litseifolius*.

### 3.2. MLT Triggers the Antioxidant Capacity in DHC Accumulation

MLT can act as a signaling molecule to maintain intracellular ROS homeostasis [11], and play a crucial role in modulating plant stress response and secondary metabolism [7]. For instance, exogenous MLT alleviates drought stress by lowering ROS and MDA levels, whereas it enhances flavonoid and carbohydrate production in *Agropyron mongolicum* [26]. In the current study, it was discovered that the H_2_O_2_ and MDA contents were both significantly decreased by MLT treatment at 100 μM (Figure 4a,b), which coincided with the optimal concentration for MLT-elicited DHC accumulation in *L. litseifolius* (Figure 2). These findings suggested that the reduction in ROS and lipid peroxidation might be closely linked to DHC metabolism in response to MLT. Furthermore, MLT initially triggered the antioxidant capacity by promoting the enzymatic activities of ROS-scavenging enzymes, as reported in cabbage and hickory [12,27]. Herein, the GR, SOD, CAT, and POD activities were enhanced significantly across all MLT treatment groups (Figure 4c–f), showing that antioxidant capacity augmentation played a crucial role in MLT-induced DHC metabolism. Additionally, only eight unigenes encoding antioxidant enzymes were found to be up-regulated by MLT, with *CAT* expression being improved most (6.33-fold), which matched the dramatically increased CAT activity (8.50-fold) in *L. litseifolius* (Figure 4e,g).

### 3.3. MLT Modulates Phytohormone Crosstalks in DHC Accumulation

The complicated crosstalks between MLT and other phytohormones are essential for MLT-mediated signaling networks [28]. Previous research has shown that MLT could enhance the plant stress response or flavonoid biosynthesis by altering the levels of major defense hormones. SA plays a crucial regulatory role to control the hypersensitive reaction to biotic and abiotic stressors. Both MLT and SA share a common precursor (chorismic acid) in their biosynthesis pathways, and they have many similarities in their functions [29]. Exogenous MLT triggered SA levels to promote innate resistance to pathogen attacks in *Arabidopsis*, while both MLT and SA had a favorable effect on total flavonoid content and antioxidant activity in bitter orange leaves [30]. Herein, MLT increased the SA amounts, while reducing conjugated SA (SAG) contents in *L. litseifolius* (Figure 5, Appendix A). These results indicated that MLT may release the conjugated types of SA into the free form, which in turn may initiate the hypersensitive signaling cascade that can lead to MLT-elicited DHC metabolism.

JA and its methyl derivative (MeJA) are the key regulators that modulate various processes, such as plant growth, development, and environmental adaption. Additionally, these phytohormones are widely recognized to be involved in stimulating the manufacture of phenylpropanoid chemicals by modulating the WD40/bHLH/MYB complex [31,32]. It was discovered that MLT treatment activated JA signaling, increased free JA and OPDA (JA precursor) contents, and resulted in higher anthocyanin levels in postharvest plums. In contrast, foliarly applied MLT significantly reduced the free JA amounts in *L. litseifolius* while enhancing the yields of MeJA and OPDA in this study (Figure 5, Appendix A), suggesting that MLT impeded the conversion of OPDA into JA. Therefore, JA might primarily function in the form of methyl ester (MeJA) in response to MLT.

As the simplest form of olefin, ET regulates the plants’ life cycle in many aspects, including stress response, root initiation, and secondary metabolism. In *Arabidopsis*, ACC (ET precursor) treatment increased the production of flavonoids in the root tips by controlling the expression of *EIN2* and *ETR1* [33]. Furthermore, the interaction between MLT and ET is commonly described. According to Xu et al. (2017) [34], MLT enhanced the accumulation of proanthocyanidins, phenols, flavonoids, and anthocyanins in grapes by modulating ET signaling, such as that of ERFs. Herein, MLT treatment induced ACC biosynthesis in *L. litseifolius* leaves (Figure 5, Appendix A). Interestingly, the expression of *EIN2* (940/f2p0/5032), *ETR1* (13983/f3p0/2719), and nine *ERFs* was also increased by MLT (Appendix A), which corresponded to the above-mentioned reports. As a result, the impact of MLT on DHC biosynthesis might be modulated by a variety of phytohormonal crosstalks, particularly in stress/secondary metabolism (SA, MeJA, OPDA, and ACC).

### 3.4. MLT Regulates Transcriptional Profiles in DHC Biosynthesis

The manipulation of transcriptional profiles is the key to increasing bioactive chemical synthesis by multiple elicitors, such as MLT, MeJA, and SA [2,12,15,16,17]. RNA-seq technology is currently applicable to investigate the transcriptome-wide regulatory network by MLT elicitation. Through RNA-seq [35], an investigation was conducted into the positive effects of MLT on cotton plants under salinity stress. They discovered that a number of up-regulated DEGs were linked to ‘flavonoid biosynthesis’, ‘photosynthesis’, ‘ROS scavenging’, and ‘hormone signaling’ [35]. In this study, RNA-seq was performed to analyze the MLT-induced transcriptional profilings between the optimal groups (MLT 100) and CK. The N50 value of RNA-seq (2692 bp, Appendix A) was substantially longer than the previous research of *L. litseifolius* [22], suggesting the higher quality of sequencing data in this research. Interestingly, the KEGG categories of ‘flavone and flavonol biosynthesis’ exhibited one of the most enriched pathways in the comparison between MLT 100 and CK (Figure 3). These results indicated that MLT may predominantly regulate the flavonoid biosynthesis in *L. litseifolius* leaves.

In general, the transcriptional regulation of diverse biological processes mainly depends on the TF families [36]. Multiple TFs can influence the biosynthesis of various flavonoids in plants, such as *PbMYB9* for anthocyanins in pear, *NtbHLH1* for proanthocyanidin in narcissus, *CsWD40* for flavan-3-ols in tea, and *TaWRKY44* for luteolin in dandelion [37,38,39,40]. In the present study, MLT enhanced the expression of multiple unigenes in *L. litseifolius* from the aforementioned TF families (Figure 6). As a result, it was hypothesized that MYBs and bHLHs might function as the key TF regulators through MLT elicitation. Among them, *MYB4* and *bHLH35* might have a positive effect on flavonoid synthesis [41,42]. Interestingly, it was also discovered that *MYB4* (42638/f2p0/900) and *bHLH35* (38525/f2p0/1344) were extensively up-regulated in MLT treatment groups (Figure 6), implying that they may essentially play a regulatory role in MLT-elicited flavonoid biosynthesis.

The pathways of DHC biosynthesis have been essentially unraveled in apple and *L. litseifilus* [2]. In general, the DHC precursor (p-coumaroyl-CoA) is derived from the phenylalanine pathway, then hydrogenated and catalyzed to generate phloretin, which is subsequently glycosylated at the 2′ or 4′ position to produce phlorizin or trilobatin, respectively [2]. Multiple rate-limiting enzymes, such as PAL, C4H, 4CL, and UGTs, participated in the catalysis of these biosynthetic processes [2]. According to our findings, MLT considerably boosted the activities of PAL, C4H, and 4CL, as well as the expression of related encoding genes in *L. litseifilus* (Figure 7). Remarkably, the strongest enzyme activities were all identified in the MLT 100 group (Figure 7), which corresponded to the results of MLT-induced DHC levels (Figure 2). Furthermore, a number of UGTs specifically regulated DHC glycosylation in apples, such as UGT71K1 (2′-O-glycosylation of phloretin to phlorizin) [20] and PGT2 (UGT88A1-like; 4′-O-glycosylation of phloretin to trilobatin) [4]. Herein, it was also found that the expression level of UGT71K1 and UGT88A1 was increased by MLT (Figure 7), which may partly account for the co-elicitation effects of MLT on phlorizin and trilobatin biosynthesis in *L. litseifilus*.

Interestingly, *MYB4* and *bHLH35* exhibited a significant co-expression with *4CL*, *C4H*, *UGT71K1*, and *UGT88A1* (Figure 10). As a result, MLT had the potential to activate the regulatory chains of ‘MYB/bHLH-*C4H*/*4CL*/*UGT88A1*-DHC biosynthesis’ in *L. litseifilus.* These results provided a potential molecular network of MLT-elicited DHC biosynthesis, which may broaden the knowledge of elicitation roles of MLT. Further research is still required to determine the molecular mechanisms underlying these regulatory pathways.

## 4. Materials and Methods

### 4.1. Plant Material

The mature *L. litseifilus* seeds were collected from their native region in Luofu Mountain, Huizhou, Guangdong Province, China. Following a 120-day period of storage in the sand, the seeds were then planted in non-woven pots (24 cm in diameter and 28 cm in height) filled with a mixture of peat, rice husk, and loess soil (3:6:1). The seeds were cultivated in a greenhouse with average temperatures of 25 °C, relative humidity of 60%, and day light length of 12 h. The seven-month-old *L. litseifilus* seedlings with a similar status of growth were chosen for further experiments.

### 4.2. MLT Elicitation Treatment

MLT (Sangon Biotech, Shanghai, China) stock solution (0.2 M) was dissolved in absolute ethyl alcohol, and diluted with distilled water to make the concentrations of 50, 100, and 200 μM. Four MLT elicitation treatments were performed, including 50 μM MLT (MLT 50), 100 μM MLT (MLT 100), 200 μM MLT (MLT 200), and water (control, CK). Each treatment group consisted of 20 plants. By using a 2 L automatic sprayer, the seedling leaves were sprayed with MLT on both sides until completely covered with water droplets. The foliar applications of MLT were performed late in the evening and conducted three times at the interval of one day. The sample leaves were harvested 14 days after the first elicitation, and promptly frozen in liquid nitrogen. They were then preserved at −80 °C for further investigation into biochemical indices, transcriptome, and qRT-PCR.

### 4.3. DHC Extraction and Quantification

The total flavonoid levels of leaf samples were then extracted and measured by the NaNO_2_-Al(NO_3_)_3_-NaOH system according to prior research [43], with rutin (Yuanye, Weifang, China) as reference. Three representative DHCs (phlorizin, trilobatin, and phloretin) were identified and quantified through HPLC analysis based on the study of Wei et al. (2020) [44], with minor modification. In brief, chromatographic analysis was performed using the Shimadzu LC-20 instrument with an SPD-M20A diode array detector, on an InertSustain C18 analytical column (250 mm × 4.6 mm, 5 µm) at 40 °C. Before measurement, the DHC extracts were dissolved in 30 mL of 70% ethanol using an ultrasonic method, and the supernatant was filtered through a 0.22 μm membrane filter. The mobile phase was gradient-eluted with acetonitrile (solvent A) and 0.04% (*v*/*v*) formic acid (solvent B). The flow rate was set at 0.8 mL min^−1^, and the detection wavelength was 285 nm. Phlorizin, trilobatin, and phloretin standards (Yuanye, China) were used as the standard calibration curves.

### 4.4. Determination of H_2_O_2_ and MDA Levels and Antioxidant Enzyme Activities

A total of 0.5 g (FW) of frozen samples was thoroughly crushed into the homogenates in 4.5 mL phosphate buffer solution (0.2 M, pH 7.4) at 4 °C. The homogenate was centrifuged to obtain the supernatant for H_2_O_2_, MDA, SOD, POD, CAT, and GR measurements, as described by the instructions of assay kits (Solarbio, Beijing, China) (Appendix A).

### 4.5. Determination of Phytohormone Levels

Samples of leaves from MLT 100 (the optimal concentration for eliciting DHC) and CK groups were collected for phytohormone quantification, including two types of SA, seven metabolites involved in JA biosynthesis, and ACC. The mixture of 0.5 g (FW) frozen samples and internal standards were extracted with a combination of 1 mL methanol/water/formic acid (15:4:1), together with internal standards. The phytohormone extracts were examined using an ultra-performance liquid chromatography–electrospray tandem mass spectrometry system, as described by Niu et al. (2014) [45]. Through the utilization of the AB Sciex QTRAP 6500 LC-MS/MS platform, the hormone levels were measured by MetWare (Wuhan, China) [46].

### 4.6. Determination of Flavonoid Biosynthesis-Related Enzyme Activities

Following the instructions, the PAL, C4H, and 4CL activities were measured by the PAL, C4H, and 4CL activity assay kits from Solarbio (Beijing, China) (Appendix A). The PAL, C4H, and 4CL spectrophotometer detection wavelengths were set to 290, 340, and 333 nm, respectively.

### 4.7. Transcriptome Sequencing Analysis

#### 4.7.1. RNA Extraction, Illumina Sequencing, and Functional Annotation

Assays were carried out for RNA extraction, RNA sequencing, and qRT-PCR on the leaf samples subjected to MLT 100 treatments (the optimal condition for DHC elicitation) and CK. Each group consisted of three biological replicates. Total RNA was extracted using the TRIzol reagent (Invitrogen, Waltham, MA, USA). Then, library construction and sequencing were performed on an Illumina Hiseq platform. Low-quality reads were removed from the raw reads. With the use of Hisat2 v2.0.5 software, the generated clean reads were then mapped into the Pacbio sequencing data obtained from our earlier research (data that were not released) [47]. Seven public databases were used for gene classification and annotation, including Nr, NT, Pfam, KOG, Swiss-Prot, KEGG, and GO databases. Coding sequences and TF predictions were performed by ANGEL and iTAK 1.7a software, respectively [48].

#### 4.7.2. DEGs and Pathway Enrichment Analysis

Gene expression levels were determined using RSEM V1.3.0 [49] and Bowtie2 V2.3.4 software [50]. The FPKM values were calculated after the read counts were normalized for each transcript. The DEG analysis of different libraries was performed using the DESeq R package [51]. Heat map plotting, correlation, and PCA were carried out by the TBTools, OmicShare (https://www.omicshare.com/tools, accessed on 23 August 2023), and Bioinformatics (https://www.bioinformatics.com.cn, accessed on 11 September 2023), respectively [52,53,54].

#### 4.7.3. qRT–PCR Validation of DEGs

The RNAprep Pure plant total RNA extraction kit (Tiangen, Beijing, China) was used to extract the total RNA from the MLT 100 and CK groups simultaneously. Then, through transcribed one-step gDNA removal and cDNA synthesis SuperMix (TransGen Biotech, Beijing, China), the cDNA was derived from the RNA samples. The SYBR Green method was utilized in the quantitative real-time PCR (qRT–PCR) assays that were carried out with Talent qPCR Premix (Tiangen, Beijing, China). The relative expression level of each gene was estimated using the 2^−ΔΔCt^ technique. Appendix A lists a comprehensive inventory of the qRT-PCR primers.

### 4.8. Statistical Analysis

The data were calculated as mean ± SDs. The results obtained from multiple treatments were compared by ANOVA before a Tukey’s post hoc test.

## 5. Conclusions

Herein, our findings indicated that the foliar application of MLT elicited the total flavonoid and DHC accumulation in *L. litseifolius*. MLT triggered the antioxidant enzyme activities, leading to a decrease in ROS levels. The biosynthesis of numerous phytohormones was improved by the MLT linked to secondary metabolism (SA, MeJA, and ACC), but free JA levels were reduced in *L. litseifolius*. Moreover, MLT enhanced the biosynthetic enzyme activities of flavonoid. RNA-seq analysis indicated that the DEGs in the antioxidant system (*SOD*, *CAT*, and *POD*), transcription factor regulation (*MYBs* and *bHLHs*), and DHC metabolism (*4CL*, *C4H*, *UGT71K1*, and *UGT88A1*) might play an essential role in MLT-induced pathways (Figure 1). Taken together, the regulatory mechanisms of MLT-elicited DHC biosynthesis were illustrated, laying a foundation for efficient DHC production in the process of *L. litseifolius* cultivation. Further practice should focus on the combined impacts of MLT-induced leaf development and DHC biosynthesis.

## Figures and Tables

**Figure 1 ijms-25-04592-f001:**
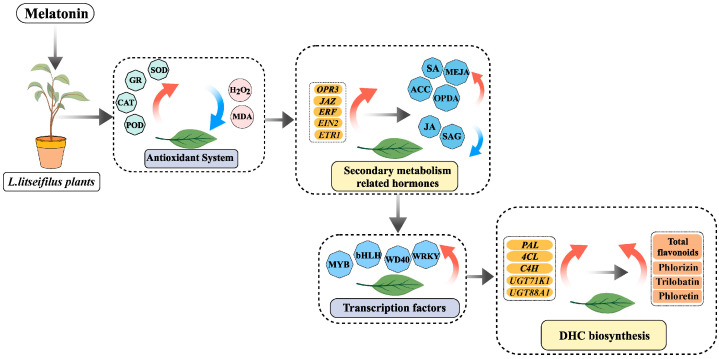
Working model of DHC biosynthesis in *L. litseifilus* induced by MLT. The red and blue arrows represented up-regulation and down-regulation of genes or metabolites, respectively.

**Figure 2 ijms-25-04592-f002:**
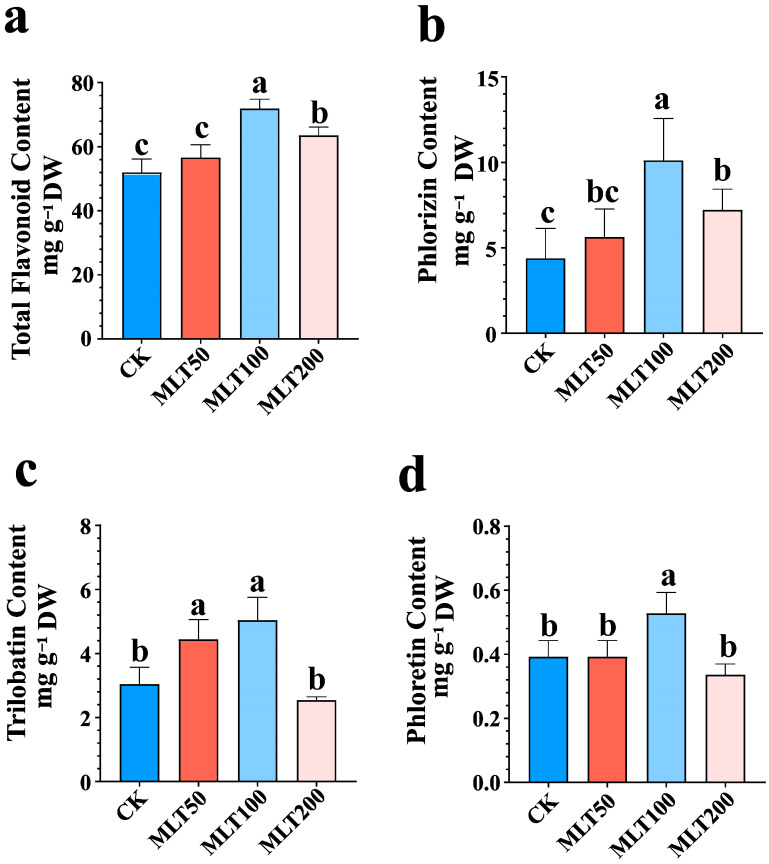
Effect of MLT on flavonoid and DHC biosynthesis in *L. litseifilus* leaves. The total flavonoid (**a**), phlorizin (**b**), trilobatin (**c**), and phloretin (**d**) in seedling leaves following 14 days of water (CK) and MLT treatments at 50 (MLT 50), 100 (MLT 100), and 200 μM (MLT 200). The values are means ± SDs. Different letters above the columns indicate significant differences (*p* < 0.05) among CK and the three MLT-treated groups.

**Figure 3 ijms-25-04592-f003:**
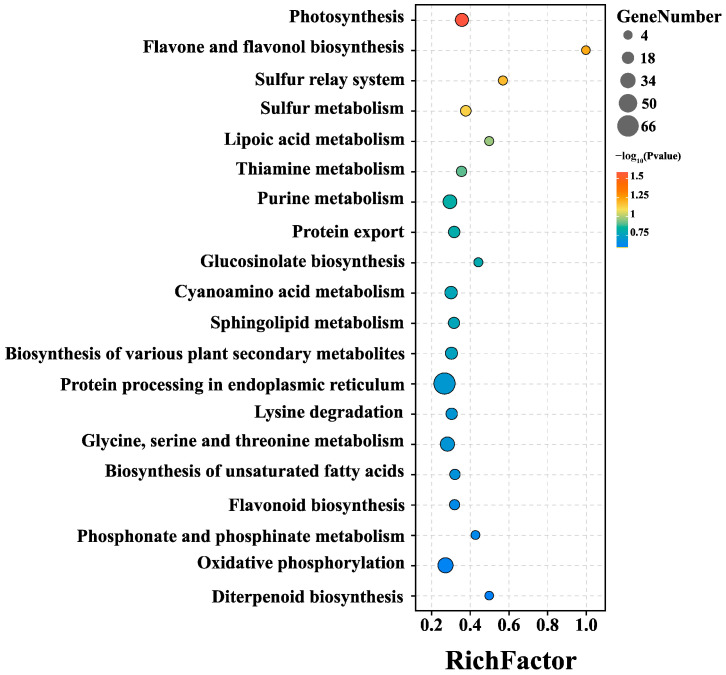
The top 20 enriched pathways of DEGs (MLT 100 vs. CK) based on the KEGG database.

**Figure 4 ijms-25-04592-f004:**
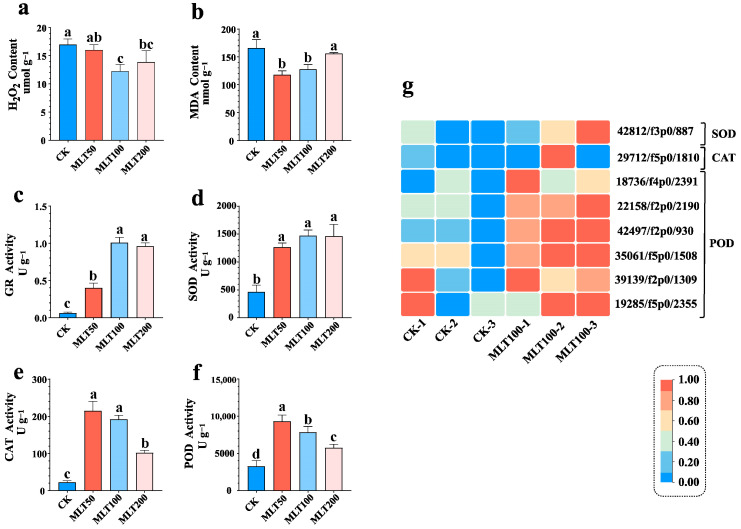
Effect of MLT on the antioxidant system and related gene expressions in *L. litseifilus*. a-f: The H2O2 (**a**) and MDA (**b**) levels, and GR (**c**), SOD (**d**), CAT (**e**), and POD (**f**) activities in seedling leaves following 14 days of water (CK) and MLT treatments at 50 (MLT 50), 100 (MLT 100), and 200 μM (MLT 200). The values are means ± SDs. Different letters above the columns indicate significant differences (*p* < 0.05) among CK and the three MLT-treated groups. (**g**) The heat map shows the up-regulated genes encoding SOD, CAT, and POD under MLT 100 treatment compared to CK. The fragment per kilobase per million (FPKM) values are log_2_ transformed.

**Figure 5 ijms-25-04592-f005:**
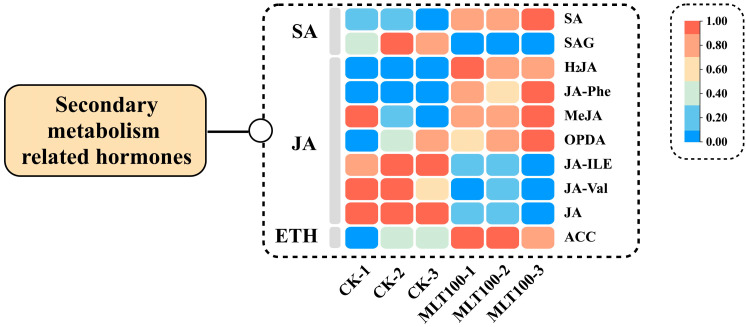
The influence of MLT on the crosstalk between phytohormones in *L. litseifilus*. The heat graph illustrates the amounts of three different types of phytohormones found in the leaves of seedlings after the treatment with water (CK) and MLT at 100 (MLT 100) for a period of 14 days.

**Figure 6 ijms-25-04592-f006:**
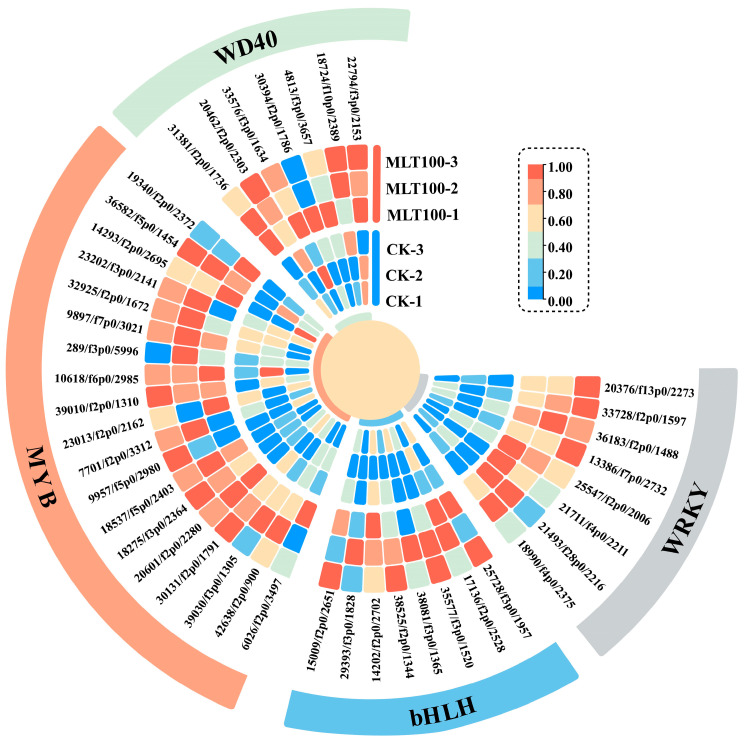
Transcriptional profiles of DEGs in TF families. The heat map reveals the up-regulated unigenes of MYB, bHLH, WD40, and WRKY families in the MLT 100 group relative to CK. The FPKM values are log_2_ transformed.

**Figure 7 ijms-25-04592-f007:**
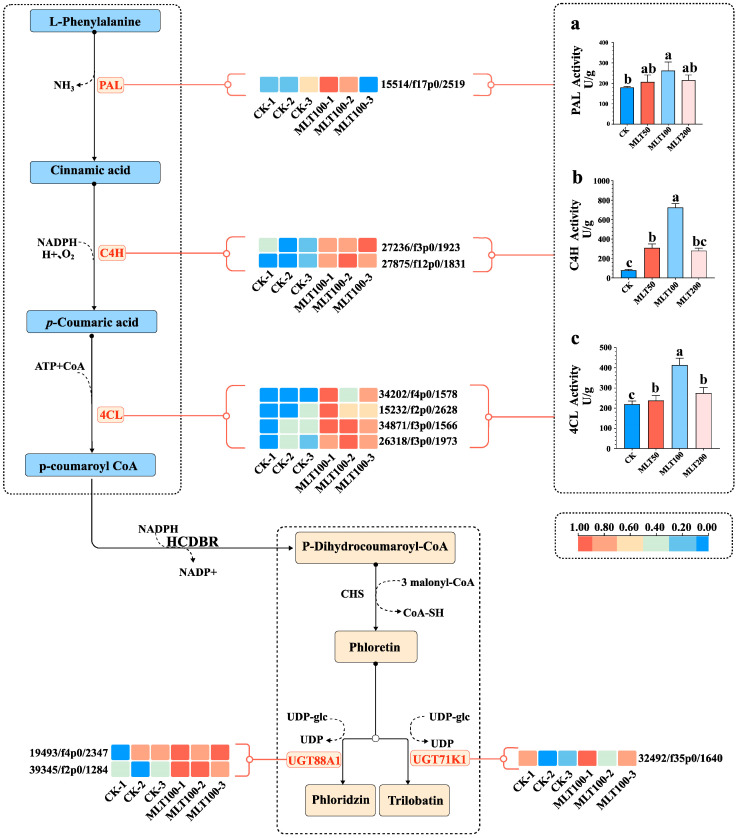
Transcriptional profiles of DEGs in TF families. The heat map revealed the up-regulated unigenes of MYB, bHLH, WD40, and WRKY families in the MLT 100 group relative to CK. The FPKM values are log_2_ transformed. (**a**–**c**): The PAL (**a**), C4H (**b**), and 4CL (**c**) activities in seedling leaves following 14 days of water (CK) and MLT treatments at 50 (MLT 50), 100 (MLT 100), and 200 μM (MLT 200). The values are means ± SDs. Different letters above the columns indicate significant differences (*p* < 0.05) among CK and the three MLT-treated groups.

**Figure 8 ijms-25-04592-f008:**
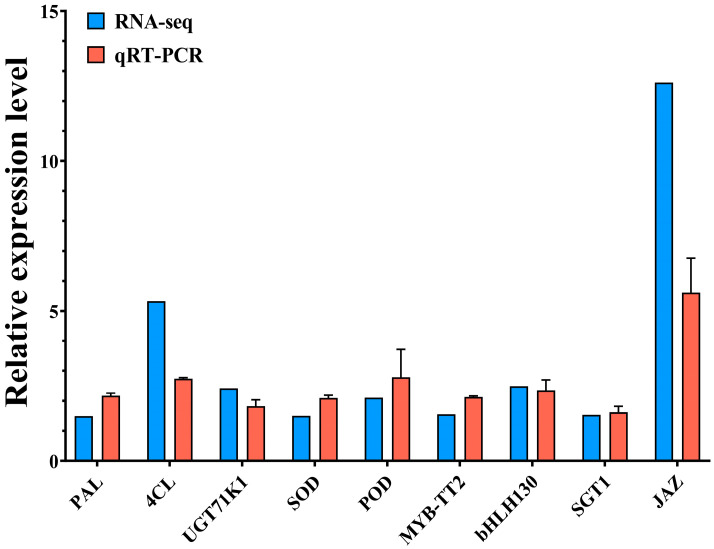
Validation of DEGs by qRT-PCR for MLT-induced flavonoid biosynthesis in *L. litseifilus*. The relative expression of nine genes from RNA-seq and qRT-PCR in a comparison between MLT100 and CK. The data are means ± SDs from three biological replicates.

**Figure 9 ijms-25-04592-f009:**
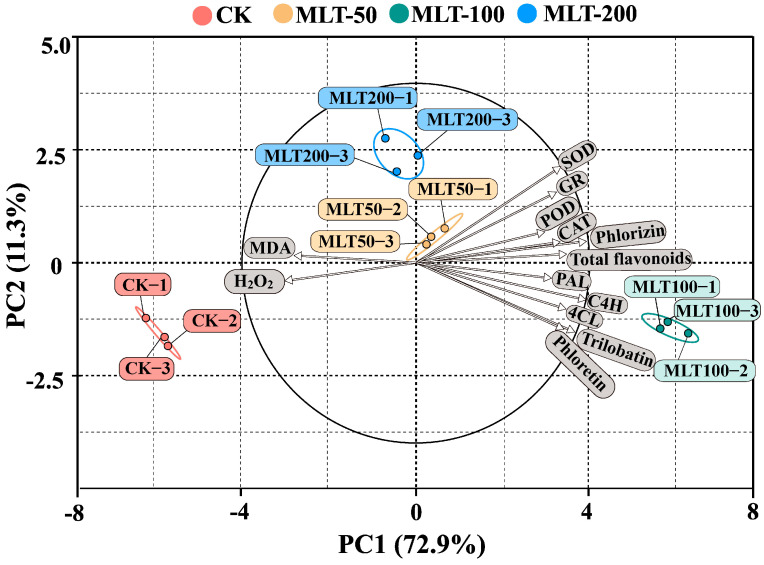
PCA among MLT treatments and physiological index attributes of *L. litseifilus*. The gray part is the physiological index.

**Figure 10 ijms-25-04592-f010:**
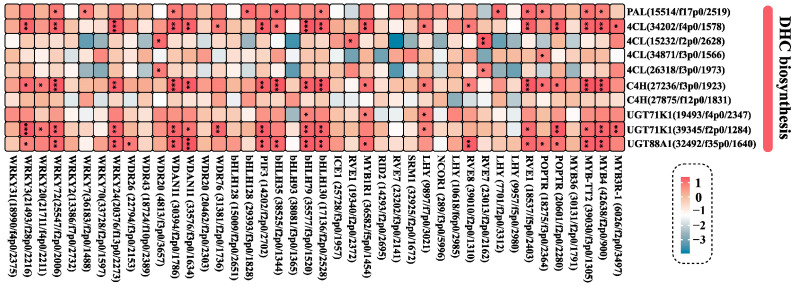
The correlation between DEGs related to TFs and potential downstream genes. The heatmap reveals the correlation coefficient and significance between the DEGs of TF (MYB, bHLH, WD40, and WRKY) and DEGs (*PAL*, *4CL*, *C4H*, *UGT71K1*, and *UGT88A1*). The color of the square represents the correlation coefficient (*r* value). *, **, and *** indicate the significant (*p* < 0.05, *p* < 0.01, and *p* < 0.001) relationship between TFs and potential downstream genes, respectively.

## Data Availability

The data will be made available upon request.

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
