# Peer review of "Exogenous Melatonin Enhances Dihydrochalcone Accumulation in Lithocarpus litseifolius Leaves via Regulating Hormonal Crosstalk and Transcriptional Profiling"

_ijms, 2024, doi:10.3390/ijms25094592_

Round 1

Reviewer 1 Report

Comments and Suggestions for Authors

The authors described the effect of foliar application of Melatonin on the total flavonoids and Dihydrochalcones accumulation in Lithocarpus litseifolius. The article is an important study but following changes/suggestions/required.

-            Check the formatting according to the journal’s instructions

-            Italics where required such as line number 473 (L. litseifolius).

-            The systematic diagram (fig.10) is suggested to be placed after or within introduction

-            Raw data  concerned to the estimation of flavonoid levels and  HPLC with reference must be shared as supplementary non publishable data of  DHCs (phlorizin, trilobatin, and  phloretin.

-            In addition to that ultra-performance liquid chromatography-electrospray  tandem mass spectrometry,  AB Sciex QTRAP 6500 LC-MS/MS platformis and PAL, C4H, and 4CL activities also desired (if possible at this stage).

-            Experimental procedure, quantification of library construction raw is recommended for verification of experiment. 

-            Once raw data has been furnished refer to the HPLC, Mass spectrometry, Deep sequencing, RNA quantification and RT PCR is more comprehensive review and cross examination of data and result is possible. 

Reviewer 2 Report

Comments and Suggestions for Authors

Dear Authors,

I have provided some suggestions and criticisms for corrections, additions, and changes within the manuscript. After the authors review these, I believe it could be deemed publishable.

Reviewer 3 Report

Comments and Suggestions for Authors

Notes and recommendations to the authors:

1. The topic of the research is current and fits within the scope of the scientific journal. It is now known that not only animals synthesize melatonin, but it is also an important plant hormone. In this aspect, the manuscript contributes to the development of Molecular Plant Sciences and more precisely to the study of the influence of exogenous melatonin on the accumulation of dihydrochalcone in the leaves of Lithocarpus litseifolius;

2. The introduction shows the development of science up to the time of conducting the study and what new the authors intend to do in the present study. At the end of the section, however, the purpose of the research is not clearly formulated, which corresponds to the title of the article, according to the requirements of the journal. Please specify the purpose of the research!

3. The results are well exposed by means of 8 figures, which are relatively successfully scaled. A principal component analysis (PCA) was performed, which further analyzed the correlations between multiple physiological indices. Please, zoom in 10-15% on fig. 6 for better visualization!

4. The Discussion section highlights the erudition and professionalism of the author team in the field of Molecular Plant Sciences. Here I recommend that the authors dig deeper into the bowels of Boosting the Production of Bioactive Compounds and how the plant sciences benefit from their research. I also recommend some reorganization of this section and fig. 9 together with the text on lines 374-379 to be moved to the Results section;

5. The Material and methods section is written relatively fully, reflecting the main activities carried out in this research. Please complete the commercial origin of the melatonin and what it was dissolved in prior to the treatments /probably an aqueous solution/? Specify the main production indicators in the greenhouse /temperature, humidity, length of light day, etc./ when growing the experimental plants!

6. In the section Conclusions to be derived from the scientific work carried out and the discussion of own and global results. The conclusions should show the scientific and applied nature of the research and be useful not only for molecular biologists and biochemists, but also for ecologists, agronomists, etc. in the plant growing industry. In this aspect, could you make some useful recommendations for the practice? If you decide to leave fig. 10 in the conclusions, please zoom in 10-15% for better visualization!

Comments on the Quality of English Language

Notes regarding the English language of the manuscript:

The manuscript is written in a fairly good scientific style, without significant errors and confusions. However, I recommend its final polishing by a professional English-speaking editor.

Round 2

Reviewer 1 Report

Comments and Suggestions for Authors

Dear Authors

The authors of the manuscript, “Exogenous melatonin enhances dihydrochalcone accumulation in Lithocarpus litseifolius leaves via regulating hormonal cross-talk and transcriptional profiling” provided all the required data in the supplementary material. but still, the diagram (figure 10) as per the claim of the authors is not placed accordingly.

Author Response

Comment 1: The diagram (figure 10) as per the claim of the authors is not placed accordingly

Response 1: Thanks for your comments. We have placed this figure after Introduction section. And the figure numbers were accordingly changed  throughout the manuscript.